# No Cost Likelihood Manipulation at Test Time for Making Better Mistakes in Deep Networks

**Shyamgopal Karthik**[*]
CVIT, IIIT Hyderabad, India

**Ameya Prabhu**
University of Oxford

**Puneet K. Dokania**
University of Oxford &
Five AI Limited

**Vineet Gandhi**
CVIT, IIIT Hyderabad, India

## Abstract

There has been increasing interest in building deep hierarchy-aware classifiers that aim to quantify and reduce the severity of mistakes, and not just reduce the number of errors. The idea is to exploit the label hierarchy (e.g., the WordNet ontology) and consider graph distances as a proxy for mistake severity. Surprisingly, on examining mistake-severity distributions of the top-1 prediction, we find that current state-of-the-art hierarchy-aware deep classifiers do not always show practical improvement over the standard cross-entropy baseline in making better mistakes. The reason for the reduction in average mistake-severity can be attributed to the increase in low-severity mistakes, which may also explain the noticeable drop in their accuracy. To this end, we use the classical Conditional Risk Minimization (CRM) framework for hierarchy aware classification. Given a cost matrix and a reliable estimate of likelihoods (obtained from a trained network), CRM simply amends mistakes at inference time; it needs no extra hyperparameters, and requires adding just a few lines of code to the standard cross-entropy baseline. It significantly outperforms the state-of-the-art and consistently obtains large reductions in the average hierarchical distance of top-$k$ predictions across datasets, with very little loss in accuracy. CRM, because of its simplicity, can be used with any off-the-shelf trained model that provides reliable likelihood estimates.

## 1 Introduction

The conventional performance measure of accuracy for image classification treats all classes other than ground truth as equally wrong. However, some mistakes may have a much higher impact than others in real-world applications. An intuitive example being an autonomous vehicle mistaking a car for a bus is a *better mistake* than mistaking a car for a lamppost. Consequently, it is essential to integrate the notion of mistake severity into classifiers and one convenient way to do so is to use a taxonomic hierarchy tree of class labels, where severity is defined by a distance on the graph (e.g., height of the Lowest Common Ancestor) between the ground truth and the predicted label (Deng et al., 2010; Zhao et al., 2011). This is similar to the problem of providing a good ranking of classes in a retrieval setting. Consider the case of an autonomous vehicle ranking classes for a thin, white, narrow band (a pole, in reality). A top-3 prediction of {pole, lamppost, tree} would be a *better* prediction than {pole, person, building}. Notice that the top-$k$ class predictions would have at least $k - 1$ incorrect predictions here, and the aim is to reduce the severity of these mistakes, measured by the average hierarchical distance of each of the top $k$ predictions from the ground truth. Silla & Freitas (2011) survey classical methods leveraging class hierarchy when designing classifiers across various application domains and illustrate clear advantages over the flat hierarchy classification, especially when the labels have a well-defined hierarchy.

There has been growing interest in the problem of deep hierarchy-aware image classification (Barz & Denzler, 2019; Bertinetto et al., 2020). These approaches seek to leverage the class hierarchy

---

[*]shyamgopal.karthik@research.iiit.ac.in

inherent in the large scale datasets (e.g., the ImageNet dataset is derived from the WordNet semantic ontology). Hierarchy is incorporated using either label embedding methods, hierarchical loss functions, or hierarchical architectures. We empirically found that these models indeed improve the ranking of the top-$k$ predicted classes – ensuring that the top alternative classes are closer in the class hierarchy. However, this improvement is observed only for $k > 1$.

While inspecting closely the top-1 predictions of these models, we observe that instead of improving the mistake severity, they simply introduce additional low-severity mistakes which in turn favours the mistake-severity metric proposed in (Bertinetto et al., 2020). This metric involves division by the number of misclassified samples, therefore, in many situations (discussed in the paper), it can prefer a model making additional low-severity mistakes over the one that does not make such mistakes. This is at odds with the intuitive notion of making better mistakes. These additional low-severity mistakes can also explain the significant drop in their top-1 accuracy compared to the vanilla cross-entropy model. We also find these models to be highly miscalibrated which further limits their practical usability.

In this work we explore a different direction for hierarchy-aware classification where we amend mistake severity at *test time* by making post-hoc corrections over the class likelihoods (e.g., softmax in the case of deep neural networks). Given a label hierarchy, we perform such amendments to the likelihood by applying the very well-known and classical approach called Conditional Risk Minimization (CRM). We found that CRM outperforms state-of-the-art deep hierarchy-aware classifiers by large margins at ranking classes with little loss in the classification accuracy. As opposed to other recent approaches, CRM does not hurt the calibration of a model as the cross-entropy likelihoods can still be used for the same. CRM is simple, requires addition of just a few lines of code to the standard cross-entropy model, does not require retraining of a network, and contains no hyperparameters whatsoever.

We would like to emphasize that we do not claim any algorithmic novelty as CRM has been well explored in the literature (Duda & Hart, 1973, Ch. 2). Almost a decade ago, Deng et al. (2010) had proposed a very similar solution using Support Vector Machine (SVM) classifier applied on handcrafted features. However, this did not result in practically useful performance because of the lack of modern machine learning tools at that time. We intend to bring this old, simple, and extremely effective approach back into the attention before we delve deeper into the sophisticated ones requiring expensive retraining of large neural networks and designing complex loss functions. Overall, our investigation into the hierarchy-aware classification makes the following contributions:

- We highlight a shortcoming in one of the metrics proposed to evaluate hierarchy-aware classification and show that it can easily be fooled and give the wrong impression of making better mistakes.

- We revisit an old post-hoc correction technique (CRM) which significantly outperforms prior art when the ranking of the predictions made by the model are considered.

- We also investigate the reliability of prior art in terms of calibration and show that these methods are severely miscalibrated, limiting their practical usefulness.

## 2 RELATED WORKS

### 2.1 COST-SENSITIVE CLASSIFICATION

Cost-sensitive classification assigns varying costs to different types of misclassification errors. The work by Abe et al. (2004) groups cost-sensitive classifiers into three main categories. The first category specifically extends one particular classification model to be cost-sensitive, such as support vector machines (Tu & Lin, 2010) or decision trees (Lomax & Vadera, 2013). The second category makes the training procedure cost-sensitive, which is typically achieved by assigning the training examples of different classes with different weights (rescaling) (Zhou & Liu, 2010) or by changing the proportions of each class while training using sampling (rebalancing) (Elkan, 2001). The third category makes the prediction procedure cost-sensitive (Domingos, 1999; Zadrozny & Elkan, 2001a). Such direct cost-sensitive decision-making is the most generic: it considers the underlying classifier as a black box and extends to any number of classes and arbitrary cost matrices. Our work comes under the third category of post-hoc amendment. We study cost-sensitive classification in

a large scale setting (e.g., ImageNet) and explore the use of a taxonomic hierarchy to obtain the misclassification costs.

## 2.2 HIERARCHY AWARE CLASSIFICATION

There is a rich literature around exploiting hierarchies to improve the task of image classification. Embedding-based methods define each class as a soft embedding vector, instead of the typical one-hot. DeViSE (Frome et al., 2013) learn a transformation over image features to maximize the cosine similarity with their respective `word2vec` label embeddings. The transformation is learned using ranking loss and places the image embeddings in a semantically meaningful space. Akata et al. (2015); Xian et al. (2016) explore variations of text embeddings, and ranking loss frameworks. Barz & Denzler (2019) project classes on a hypersphere, such that the correlation of class embeddings equals the semantic similarity of the classes. The semantic similarity is derived from the height of the lowest common ancestor (LCA) in a given hierarchy tree.

Another line of work directly alters the loss functions or the algorithms/architectures. Zhao et al. (2011) propose a weighted (hierarchy-aware) multi-class logistic regression formulation. Verma et al. (2012) optimize a context-sensitive loss to learn a separate distance metric for each node in the class taxonomy tree. Wu et al. (2016) combine losses at different hierarchies of the tree by learning separate, fully connected layers for each level post a shared feature space. Bilal et al. (2017) add branches at different depths of AlexNet architecture to fuse losses at different levels of the hierarchy. Brust & Denzler (2019) use conditional probability chains to derive a novel label encoding and a corresponding loss function.

Most deep learning-based methods overlook the severity of mistakes, and the evaluation revolves around counting the top-$k$ errors. Bertinetto et al. (2020) has revived the interest in this direction by jointly analyzing the top-$k$ accuracies with the severity of errors. They propose two modifications to cross-entropy to better capture the hierarchy: one based on label embeddings (Soft-labels) and the other, which factors the cross-entropy loss into the individual terms for each of the edges in the hierarchy tree and assigns different weights to them (Hierarchical cross-entropy or HXE).

Our method uses models trained with vanilla cross-entropy loss and alters the decision rule to pick the class that minimizes the conditional risk where the condition is being imposed using the known class-hierarchy. On similar lines, Deng et al. (2010) study the effect of minimizing conditional risk on the mean hierarchical cost. They leverage the ImageNet hierarchy for cost and compute posteriors by fitting a sigmoid function to the SVM's output or taking the percent of neighbours from a class for Nearest Neighbour classification. Our work investigates the relevance of CRM in the deep learning era and highlights the importance of looking beyond mean hierarchical costs and jointly analyzing the role of accuracy and calibration.

## 2.3 CALIBRATION OF DEEP NEURAL NETWORKS

Networks are said to be well-calibrated if their predicted probability estimates are representative of the true correctness likelihood. Calibrated confidence estimates are important for model interpretability and its use in downstream applications. Platt scaling (Platt et al., 1999), Histogram binning (Zadrozny & Elkan, 2001b) and Isotonic regression (Zadrozny & Elkan, 2002) are three common calibration methods. Although originally proposed for the SVM classifier, their variations are used in improving the calibration of neural networks (Guo et al., 2017). Calibrated probability estimates are particularly important when cost-sensitive decisions are to be made (Zadrozny & Elkan, 2001b) and are often measured using Expected Calibration Error (ECE) and Maximum Calibration Error (MCE) (Niculescu-Mizil & Caruana, 2005; Naeini et al., 2015; Mukhoti et al., 2020).

We desire models with high accuracy that have low calibration error and make less severe mistakes. However, there is often a compromise. Studies in cost-sensitive classification (Jan et al., 2012) reveal a trade-off between costs and error rates. Reliability literature aims to obtain better calibrated deep networks while retaining top-$k$ accuracy (Seo et al., 2019). We further observe that methods like Soft-labels or Hierarchical cross-entropy successfully minimize the average top-$k$ hierarchical cost, but result in poorly calibrated networks. In contrast, the proposed framework retains top-$k$ accuracy and good calibration, while significantly reducing the hierarchical cost.

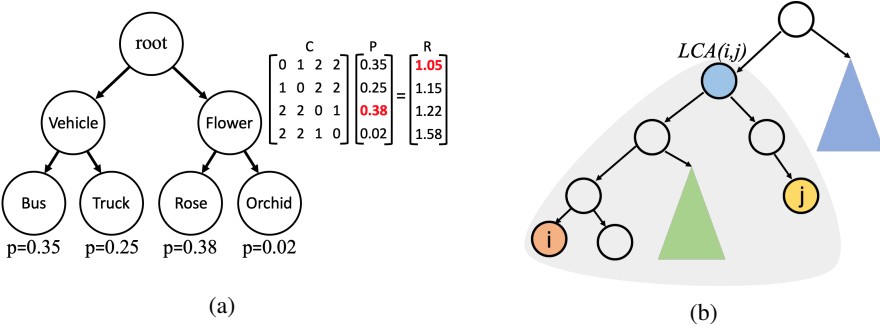

Figure 1: (a) Consider a four class hierarchy tree and corresponding leaf predictions obtained using a cross-entropy baseline. The risk computation is shown beside the tree. $\operatorname{argmax} p(\mathbf{y}|\mathbf{x})$ predicts the class "rose", while the $\operatorname{argmin} R(\mathbf{y} = k|\mathbf{x})$ predicts the class "bus". (b) Two nodes $i$ and $j$ and the subtree (shaded gray) originating at their lowest common ancestor $LCA(i, j)$.

## 3  APPROACH

The $K$-class classification problem comes with a training set $S = \{(\mathbf{x}_i, \mathbf{y}_i)\}_{i=1}^N$, where label $\mathbf{y}_i \in \mathcal{Y} = \{1, 2, ..., K\}$. The classifier is a deep neural network $f_\theta : \mathcal{X} \to p(\mathcal{Y})$ parametrized by $\theta$ which maps the input samples to a probability distribution over the label space $\mathcal{Y}$. The $p(\mathbf{y}|\mathbf{x})$ is typically derived using a softmax function on the logits obtained for an input $\mathbf{x}$. Given $p(\mathbf{y}|\mathbf{x})$, the network minimizes cross-entropy with the ground truth class over samples from the training set, and uses SGD to optimize $\theta$, forming the standard hierarchy-agnostic cross-entropy baseline. The decision rule is naturally given by $\operatorname{argmax}_k p(\mathbf{y} = k|\mathbf{x})$.

The classical CRM framework (Duda & Hart, 1973) can be adapted to image classification by taking the trained model with a given $\theta$ and incorporating the hierarchy information at deployment time. A symmetric class-relationship matrix $\mathbf{C}$ is created using the given hierarchy tree (which can either be drawn from the WordNet ontology or an application specific taxonomy), where $\mathbf{C}_{i,j}$ is the height of the lowest common ancestor $LCA(\mathbf{y}_i, \mathbf{y}_j)$ between classes $i$ and $j$. The height of a node is defined as the number of edges between the given node and the furthest leaf. $\mathbf{C}_{i,j}$ is zero when $i = j$ and is bounded by the maximum height of the hierarchy tree.

Given an input $\mathbf{x}$, the likelihood $p(\mathbf{y}|\mathbf{x})$ is obtained by passing the sample through the network $f_\theta(\mathbf{x})$. The only modification we make is in the decision rule, which now selects the class that minimizes the conditional risk $R(\mathbf{y} = k|\mathbf{x})$, given by:

$$\operatorname*{argmin}_k R(\mathbf{y} = k|\mathbf{x}) = \operatorname*{argmin}_k \sum_{j=1}^K \mathbf{C}_{k,j} \cdot p(\mathbf{y} = j|\mathbf{x}) \tag{1}$$

For the ease of the reader, we illustrate a four-class example in Figure 1a, comparing predictions obtained using the standard cross-entropy baseline (leaf nodes), and the prediction using CRM (Eq. (1)) for a given class-relationship matrix. Given the probability of each class $p(\mathbf{y}|\mathbf{x})$, $\operatorname{argmin} R(\mathbf{y}|\mathbf{x})$ is the Bayes optimal prediction. It is guaranteed to achieve the lowest possible overall cost, i.e. lowest expected cost over all possible examples weighted by their probabilities (Duda & Hart, 1973, Ch. 2).

Depending on the cost-matrix and $p(\mathbf{y}|\mathbf{x})$, the top-1 prediction of the CRM applied on cross-entropy might differ from the top-1 prediction of the cross-entropy baseline. However, because of the over-confident nature of recent deep neural networks, we observe that the top-1 probability of $p(\mathbf{y}|\mathbf{x})$ is greater than 0.5 for significant number of test samples. Below we prove that in such situations where $\max p(\mathbf{y}|\mathbf{x})$ is higher than the sum of other probabilities, the post-hoc correction (CRM) does not change the top-1 prediction irrespective of the structure of the tree. Since the second highest probability is guaranteed to be less than 0.5 by definition, our correction can effectively re-rank the classes. Experimentally we find it to significantly reduce the hierarchical distance@$k$.

**Theorem 1.** *If* $\max(p(\mathbf{y}|\mathbf{x})) > 0.5$*, then* $\operatorname{argmin}_i \sum_{j=1}^K \mathbf{C}_{i,j} \cdot p(\mathbf{y} = j|\mathbf{x})$ *and* $\operatorname{argmax} p(\mathbf{y}|\mathbf{x})$ *are identical irrespective of the tree structure and both lead to the same top-1 prediction.*

*Proof.* Consider the tree illustrated in Figure 1b; two leaf nodes (class labels) $i,j$ and the subtree ($T_{ij}$) rooted at their Lowest Common Ancestor. Assuming the height of the $LCA(i,j) = h$ and $\text{argmax } p(\mathbf{y}|\mathbf{x}) = i$, the risk $R(\mathbf{y} = j|\mathbf{x}) = R(j)$ is given as:

$$R(j) = h \cdot p(i) + \sum_{k \in T_{ij} \setminus \{i\}} \mathbf{C}_{j,k} \cdot p(k) + \sum_{\forall k \notin T_{ij}} \mathbf{C}_{j,k} \cdot p(k)$$

Ignoring the cost of other nodes inside $T_{ij}$, we get $R(j) \geq h \cdot p(i) + \sum_{\forall k \notin T_{ij}} \mathbf{C}_{j,k} \cdot p(k)$. Similarly, for the risk of class $i$:

$$R(i) \leq h \cdot (1 - p(i)) + \sum_{\forall k \notin T_{ij}} \mathbf{C}_{i,k} \cdot p(k)$$

Outside the subtree rooted at $T_{i,j}$, $\mathbf{C}_{i,k} = \mathbf{C}_{j,k} \forall k$ and therefore without loss of generality we can say that $R(i) < R(j)$, if $p(i) > 0.5$. □

## 4 EXPERIMENTS

We evaluate our method on two large-scale hierarchy-aware benchmarks: (i) tieredImageNet-H for a broad range of classes and (ii) iNaturalist-H for fine-grained classification, both of which are complex enough to cover a large number of visual concepts. We closely follow the experimental pipeline from Bertinetto et al. (2020) including the train/validation/test splits, hyperparameters for training models, and evaluation metrics.

**Experimental Details:** All models are trained using a ResNet-18 architecture (pre-trained on ImageNet) using an Adam optimizer for 200K updates using a mini-batch of 256 samples, a learning rate of $10^{-5}$, and standard data augmentation of flips and randomly resized crops. We train all the hierarchy-aware models – Hierarchical cross-entropy (HXE) (Bertinetto et al., 2020), Soft-labels (Bertinetto et al., 2020), YOLO-v2 (Redmon & Farhadi, 2017), DeViSE (Frome et al., 2013), and Barz & Denzler (2019) – along with a cross-entropy baseline. We pick the epoch corresponding to the lowest loss on the validation set along with two epochs preceding and succeeding it and report the average of the results obtained from these five checkpoints on the test set. Unlike Bertinetto et al. (2020) we do not preprocess the dataset to downsample the images to 224×224 as it noticeably reduces the accuracy. Instead, we use the RandomResizedCrop() augmentation to crop the images to a 224×224 resolution. This accounts for a small, but significant improvement in performance across models, thus leading to stronger baselines.

**Metrics:** We primarily focus on two major metrics: (i) top-1 error, and (ii) average hierarchical distance@$k$, which is the mean LCA height between the ground truth and each of the $k$ most likely classes. These metrics capture different views of the problem: top-1 error treats all classifier mistakes the same, whereas average hierarchical distance@1 captures a notion of mistake severity, i.e., *better* or *worse* mistakes. Average hierarchical distance@$k$ captures the notion of ranking/ordering the predicted classes closer to the ground truth class. This metric, also used in Bertinetto et al. (2020), is a natural extension of the hierarchical distance@1 proposed by Russakovsky et al. (2015) in the original ImageNet evaluation. We also investigate the average mistake-severity metric suggested in Bertinetto et al. (2020) which computes the hierarchical distance between the top-1 prediction and the ground truth for all the *misclassified* samples. Note that LCA is a log-scaled distance: an increment of 1.0 signifies an error of an entire level of the tree. In the simple case of a full binary tree, an increase by one level implies that the number of possible leaf nodes doubles.

### 4.1 HIERARCHICAL DISTANCE OF TOP-1 PREDICTIONS

Hierarchy-aware classification methods typically seek to make better mistakes (less costly in terms of hierarchical distance). It is essential that the evaluation metric correctly measures this goal, i.e. a higher value of the evaluation metric should reflect that the model indeed makes better mistakes. Below we discuss a shortcoming of the average *mistake-severity metric* proposed in Bertinetto et al. (2020) which considers the mistake severity averaged only over the *incorrectly* classified samples,

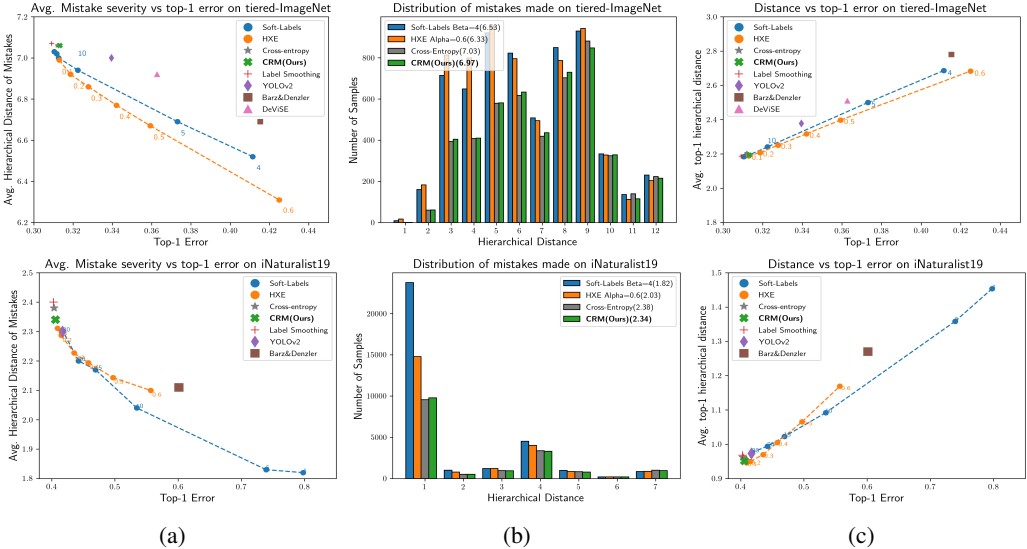

Figure 2: (a) Trade-off between hierarchical distance@1 and the top-1 error. The mean here is taken only over the misclassified samples. (b) Histogram of the severity of top-1 mistakes (height of the LCA). The number in the bracket is the mean mistake severity. (c) Trade-off between hierarchical distance@1 and the top-1 error. The mean is taken over all the test samples (the hierarchical distance@1 is zero for correctly classified samples). The top and bottom row correspond to tiered-ImageNet and iNaturalist19 datasets respectively.

and show that it can be misleading in the sense that a model can show improved performance over this metric, while just making additional low-severity mistakes.

In Figure 2a, we evaluate different approaches only on the set of incorrectly classified samples (hence *different* test sets for different models as the mistakes will be different). It seems to indicate that recently proposed methods are able to achieve a good trade-off between top-1 error and mistake severity. We select models that show a marked trade-off in terms of the mistake-severity metric – Soft-labels with $\beta = 4$ and HXE with $\alpha = 0.6$ – and analyze the frequency of mistakes at different levels of severity (illustrated in Figure 2b). Surprisingly, we observe that in these regimes, HXE and Soft-labels largely do not make better mistakes; they mostly make additional low-severity mistakes. This behaviour is better demonstrated in the histograms shown in Appendix A.1 (Figure 4). For example, in the case of Soft-labels on iNaturalist19 dataset, it is evident from Figure 4 that as $\beta$ decreases, the number of less-severe mistakes increases, whereas, the high-severity mistakes remain more or less the same. Similar observations can be made for HXE. This behaviour is not captured in Figure 2a as the metric here involves division by the number of mistakes made by the model. More precisely, say the high severity mistakes made by two models are exactly the same ($d_h > 0$) over the same number of mistakes ($m > 0$). Now, if the second model makes additional $n > 0$ mistakes with overall distance severity of $d_l > 0$, then it is straightforward to observe that $\frac{d_h}{m} \geq \frac{d_h + d_l}{m+n}$ if $\frac{d_h}{m} \geq \frac{d_l}{n}$. This implies that the metric would *prefer a model making additional low-severity mistakes* as long as the impact of the severity due to these additional mistakes is less than the overall impact by the high-severity ones.

We avoided this shortcoming by using the hierarchical distance@1 computed over *all* the samples (Russakovsky et al., 2015). As shown in Figure 2c, the best-performing ones in Figure 2a now show the highest hierarchical distance@1 as we account for the additional number of low-severity mistakes made by them. Note, distance@1 was also used in Bertinetto et al. (2020), however, they also proposed the above mentioned mistake-severity metric and performed analyses over it, which, as discussed and showed empirically, can easily mislead us towards choosing a classifier that just makes additional low severity mistakes while not improving the overall mistake severity at all.

In this more reliable evaluation set-up, we observe CRM (ours) marginally reduces mistake severity compared to cross-entropy. We would like to emphasize that cross-entropy provides near best results. Overall, our experiments suggest that existing methods reduce the average mistake-severity metric by largely making additional low-severity mistakes. This also explains why such models provide lower test accuracy (top-1). Resolving this issue, we see that no prior art significantly out-

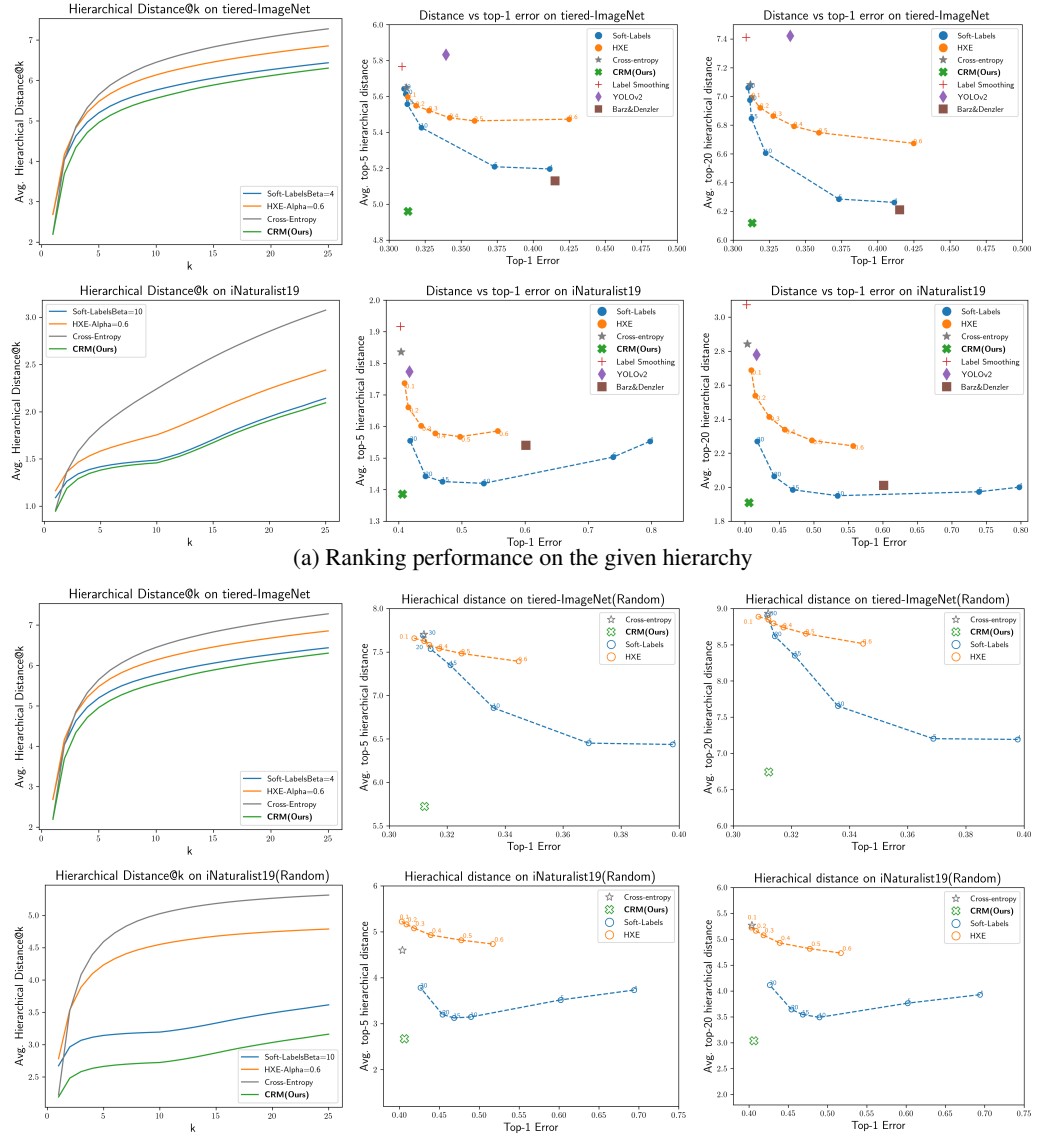

(a) Ranking performance on the given hierarchy

(b) Ranking performance on a randomly class-shuffled hierarchy.

Figure 3: Left: Average hierarchical distance@$k$ with varying $k$. Middle: Average hierarchical distance@5 vs top-1 error. Right: Average hierarchical distance@20 vs top-1 error.

performs the cross-entropy baseline either in making better mistakes (distance@1) or in the top-1 accuracy.

## 4.2 HIERARCHICAL DISTANCE OF TOP-$k$ PREDICTIONS

We now compare the ordering of classes provided by each of these classifiers. Ranking predictions give us significant insight into how reliably the predictions align with the hierarchy. We measure the quality of ranking using the average hierarchical distance@$k$, for various values of $k$ and present them in Figure 3a (left). We find that CRM significantly outperforms all the competing methods, giving the best hierarchically aligned models on the hierarchical distance@$k$. Note, for $k > 1$, recent approaches also provide improvement in distance@$k$ compared to the cross-entropy model.

A better ranking of classes often comes with a significant trade-off with top-1 accuracy. We plot the hierarchical distance@$k$ with top-1 accuracy for $k = 5$ and $k = 20$ in Figure 3a (middle and right) to better understand this trade-off. Interestingly, we observe that CRM improves ranking with almost no loss in top-1 performance and outperforms other methods by a substantial margin.

| Loss Function | tieredImageNet-H | | | | iNaturalist-H | | | |
| --- | --- | --- | --- | --- | --- | --- | --- | --- |
| | ECE | | MCE | | ECE | | MCE | |
| | pre T | post T | pre T | post T | pre T | post T | pre T | post T |
| Soft-labels ($\beta$=15) | 7.05% | 4.79% | 18.55% | 13.03% | 34.64% | 16.63% | 54.86% | 26.87& |
| Soft-labels ($\beta$=10) | 29.55% | 6.36% | 39.95% | 22.07% | 29.55% | 19.87% | 39.95% | 33.29% |
| Soft-labels( $\beta$=5) | 58.99% | 10.92% | 83.53% | 26.86% | 24.53% | 17.20 | 88.32% | 55.68% |
| Soft-labels ($\beta$=4) | 57.16% | 11.12% | 86.92% | 27.44% | 19.29% | 11.46% | 19.29% | 56.06% |
| HXE ($\alpha$=0.2) | 1.53% | 1.53% | 5.84% | 5.84% | 4.37% | 1.50% | 7.73% | 3.61% |
| HXE ($\alpha$=0.4) | 2.44% | 2.44% | 5.48% | 5.48% | 1.13% | 1.13% | 2.62% | 2.62% |
| HXE($\alpha$=0.5) | 3.95% | 2.61% | 7.84% | 5.40% | 2.46% | 2.46% | 6.77% | 6.77% |
| HXE ($\alpha$=0.6) | 6.25% | 3.28% | 10.75% | 6.58% | 5.24% | 5.24% | 11.20% | 11.20% |
| Label-Smoothing | 9.61% | 2.33% | 15.43% | 6.13% | 4.93% | 1.11% | 7.35% | 3.35% |
| Cross-Entropy | 1.61% | 1.61% | 4.27% | 4.27% | 4.32% | 1.42% | 8.18% | 3.26% |

Table 1: ECE and MCE for the various models on the tiered-ImageNet and iNaturalist19 datasets before and after temperature scaling. The optimal temperature was found on the validation set and results reported on the test set.

An interesting extension is to analyze how dependent these approaches are on a given hierarchy, and how modifying the hierarchy might impact their behaviour. To test this, we randomly shuffle the classes at the leaf nodes of a given tree structure and compare ranking performance in Figure 3b. We observe that even though CRM does not explicitly use the hierarchy while training, it provides drastic reduction in the hierarchical distance@$k$ compared to all the previous methods. High accuracy of CRM in this case is because of the fact that it is post-hoc and for highly confident models such as deep networks, its top-1 accuracy remains largely unchanged (refer Theorem 1). On the other hand, models depending on the tree-structure during training (directly or indirectly) will try to fit to the structure, which can be harmful in situations where the tree structure is not very reliable. For example, if the tree structure implies that 'cat' is closer to 'person' than a 'dog', then the models incorporating such information while learning the feature space might not be able to learn a robust classifier and might potentially end-up making more mistakes, as also validated in our experiments.

### 4.3 IMPACT OF LABEL HIERARCHY ON THE RELIABILITY

In order for models to be useful in safety-critical scenarios, they should be calibrated so that they are not wrong with high confidence. To this end, we analyse the reliability of the output probabilities of Softlabels, HXE, label smoothing, and CRM (which is the vanilla cross-entropy likelihood) using widely accepted metrics such as ECE (Expected Calibration Error) and MCE (Maximum Calibration Error) in Table 1. Softlabels and HXE, for example, show clear trends of increasing degradation in calibration on better class ranking (as measured by distance@$k$), i.e., *the more they attempt to adhere to the hierarchy, the less reliable their probability estimates become*. We additionally experiment with improving calibration in all the above models using temperature scaling. We observe that it reduces miscalibration as measured by the ECE and MCE scores, but most models still remain far worse than the cross-entropy baseline. Changes in ECE/MCE were unnoticeable when using the probability estimates corresponding to CRM predictions (taking $p(\mathbf{y}|\mathbf{x})$ corresponding to $\operatorname{argmin} R(\mathbf{y}|\mathbf{x})$) instead of maximum cross-entropy prediction.

These experiments clearly suggest that while the focus should turn into developing models that make better mistakes, we should also make sure that such models are reliable by understanding how incorporating the label hierarchy during training might impact the likelihood estimates.

## 5 CONCLUSION

We proposed using Conditional Risk Minimization (CRM) as a tool to amend likelihood in a post-hoc fashion to obtain hierarchy-aware classifiers, an approach that is different from the three dominant paradigms: hierarchy-aware losses, hierarchy-aware architectures, and label embedding methods. We illustrated an issue with the mistake-severity metric that, otherwise, could give a wrong impression of improvement while the model might just be making additional mistakes to fool the metric.

In terms of better ranking predictions, our proposed post-hoc correction consistently outperforms state-of-the-art methods in deep hierarchy-aware image classification by large margins in terms of decrease in hierarchical distance@$k$, with little to no loss in top-1 accuracy. We find the direction of post-hoc corrections promising as it can simultaneously deliver calibration, accuracy, and better class ranking efficiently with surprisingly little trade-offs in either.

Overall, the literature on hierarchy-aware image classification has shown the WordNet hierarchy's effectiveness in improving performance. However, previous works assumed that all the edges in the tree are equally important. A future avenue for exploration would be to learn the weights of these edges in order to compute a more effective measure of mistake severity.

## ACKNOWLEDGEMENTS

SGK and AP would like to thank Saujas Vaduguru, Aurobindo Munagala and Arjun P for detailed feedback on the manuscript. PKD would like to thank Sina Samangooei for constructive comments. This work was supported by the following grants/organisations: Early Career Research Award, ECR/2017/001242, from Science and Engineering Research Board (SERB), Department of Science & Technology, Government of India; EPSRC/MURI grant EP/N019474/1; Facebook (DeepFakes grant); and Five AI Ltd., UK.

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

# A  APPENDIX

## A.1  VARIATION OF MISTAKE SEVERITY WITH HIERARCHY ALIGNMENT

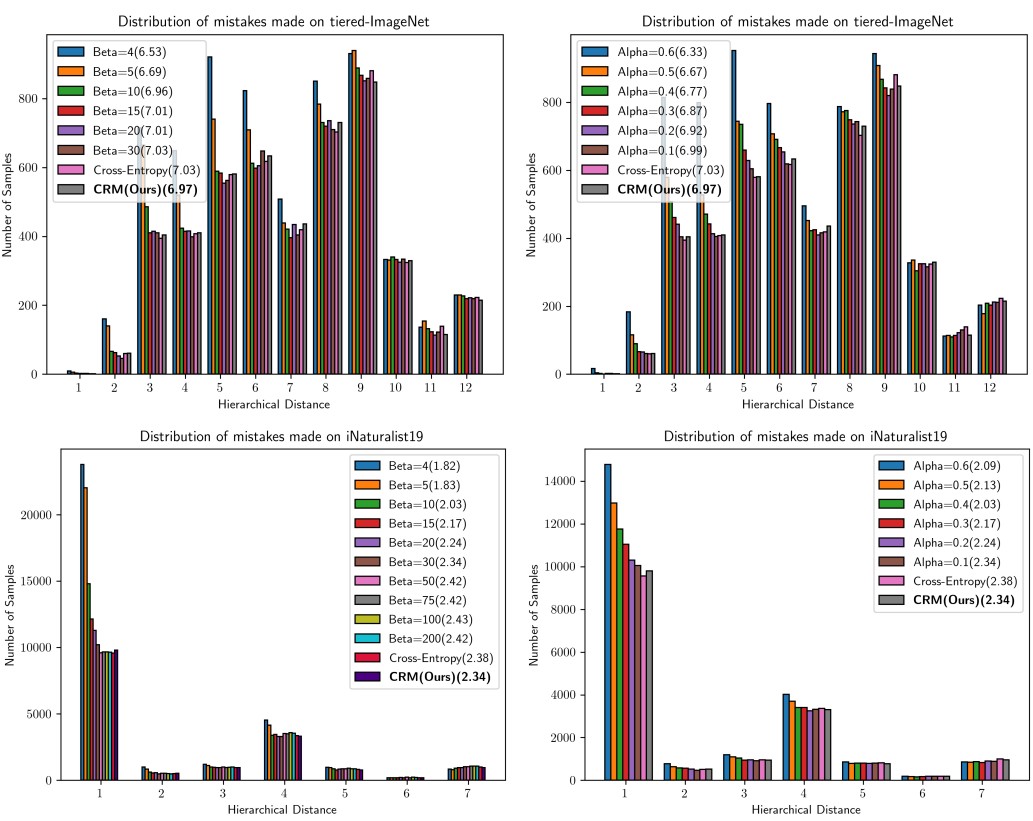

Figure 4: Mistake distribution for all values of Softlabels across $\beta$ (left) and HXE across $\alpha$ (right) as well baselines including cross-entropy and CRM for reference across tieredImagenet-H (top) and iNaturalist-19 (bottom).

We first analyze the change of the distribution of mistakes across Softlabels (left column) and HXE (right column) models in Figure 4 with results in tieredImageNet-H (top row) and iNaturalist-H (bottom row). As we try to align the model better with the hierarchy by decreasing $\beta$ and increasing $\alpha$, we observe the same trends with better alignment to hierarchy – their mistakes are equally bad compared to cross-entropy near the right end (high severity), and they increasingly make more mistakes in the left end (lower severity), lowering the average over mistakes but not always making better mistakes. This gives evidence that the models have the tendency to decrease their mistake severity (shown in the legend) by largely making lots of additional mistakes and *not making better mistakes* in the large $\beta$ and small $\alpha$ regimes.

## A.2  VARIATION OF CLASS RANKING WITH HYPERPARAMETERS

We similarly analyze the variation of ranking classes measured by average hierarchical distance@$k$ for Softlabels and HXE with different hyperparameters. We present our results in Figure 5, by varying Softlabel with different $\beta$ values on the left and HXE with different $\alpha$ values on the right. We observe that the previously chosen values $\beta = 4$ and $\alpha = 0.6$ perform the best in ranking in all cases except $\beta = 4$ in iNaturalist where $\beta = 10$ performs the best, which we updated in Figure 3. We observe there that CRM still outperforms these methods by large margins.

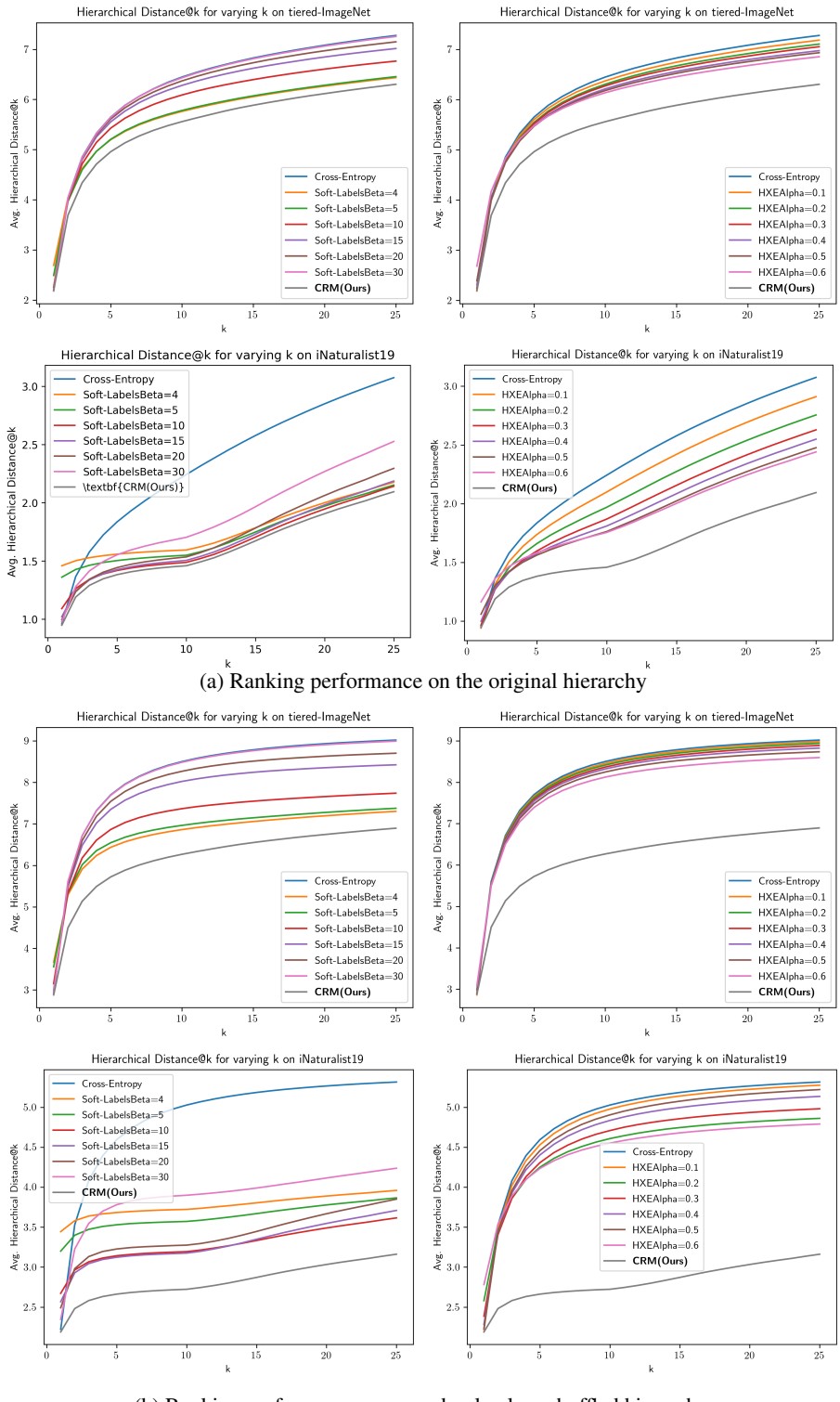

(a) Ranking performance on the original hierarchy

(b) Ranking performance on a randomly class-shuffled hierarchy.

Figure 5: Average hierarchical distance@$k$ with varying k across different hyperparameters of Soft-labels (left) and HXE (right). We observe our selected hyperparameters (alpha=0.6) and (beta=4) perform the best among others.

## A.3 Variation of Calibration Across Hierarchy

We can additionally calculate ECE across levels in the hierarchy by sequentially shrinking leaf nodes from the maximum depth (corresponding to flat classification). The ECE at depth $i$ (from root

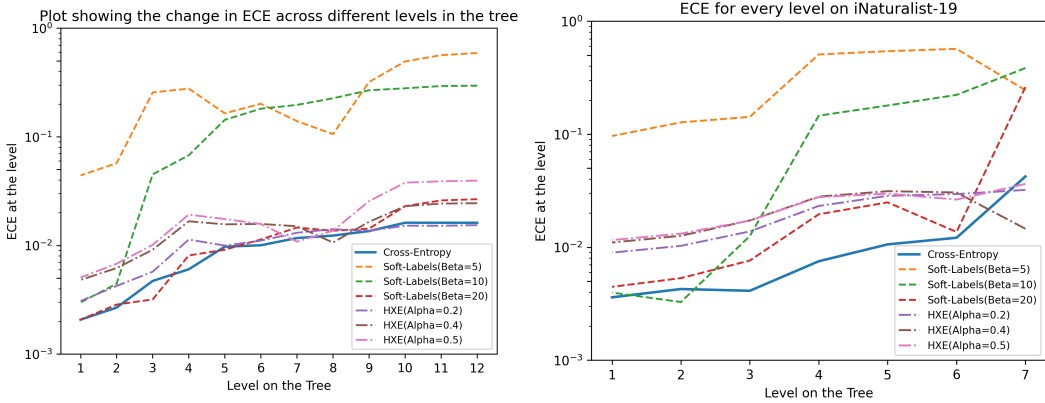

Figure 6: Calibration across various levels in the hierarchy for different models

assigned depth 0) is defined as obtaining probabilities for nodes with most depth which are at most at level $i$. Their probabilities are obtained by summing up probabilities of their children nodes (level $> i$). We present the results in Figure 6, where we observe that overall the calibration increases as you go up the hierarchy. The cross-entropy baseline shows a consistent decreasing trend, but the other models have aberrations where the calibration error increases going up the hierarchy, especially the models with high calibration errors.

## A.4 INCREASING BETA FOR SOFT-LABELS ON INATURALIST19

We also experiment with increasing $\beta$ for Soft-labels on the iNaturalist19 dataset by trying the values of 50, 75, 100 and 200 respectively. These results are shown in Table 2. We can see that increasing Beta does not significantly improve top-1 accuracy while worsening the hierarchical distance metrics. We additionally observe this in Figure 2, where as we increase $\beta$ the graph shoots up with little leftward shift.

| Model | Accuracy | Hier. Distance@1 | Hier. Distance@5 | Hier. Distance@20 |
|---|---|---|---|---|
| Soft-labels(Beta=30) | 0.5819± 0.001 | 0.976± 0.006 | 1.554±0.006 | 2.267±0.004 |
| Soft-labels(Beta=50) | 0.5880 ± 0.003 | 1.003 ± 0.01 | 1.879± 0.003 | 2.823±0.011 |
| Soft-labels(Beta=75) | 0.5876±0.002 | 0.997±0.005 | 1.916±0.005 | 2.909±0.017 |
| Soft-labels(Beta=100) | 0.585±0.001 | 1.01±0.004 | 1.926±0.005 | 2.93±0.014 |
| Soft-labels(Beta=200) | 0.5877±0.004 | 0.999±0.009 | 1.924±0.005 | 2.935±0.012 |
| Cross-Entropy | 0.5962±0.002 | 0.96±0.004 | 1.836±0.003 | 2.841±0.01 |

Table 2: Increasing Beta for Soft-labels on iNaturalist19 beyond 30 does not increase accuracy while significantly worsening the hierarchical distance metrics.

