# OpenReview forum: "No Cost Likelihood Manipulation at Test Time for Making Better Mistakes in Deep Networks"
_ICLR.cc/2021/Conference — ICLR 2021 Poster_

### Official Review · AnonReviewer3 · 2020-10-15
**Promising, simple method but flawed experiments/arguments**

**Rating:** 6
**Confidence:** 3

**Review:**

This paper addresses the problem of hierarchy-aware classification, which utilizes a hierarchy to specify certain mistakes as being worse than others. They make two main contributions.

First, they claim that the metric used in prior work, average mistake severity, is flawed because it rewards methods that make many "easy" mistakes, as opposed to fewer, "harder" mistakes. Based on this analysis, they claim that no prior methods actually improve over the simplest baseline of cross-entropy (i.e. doing nothing).

Second, they introduce their own method, an adaptation of the classical CRM framework. They argue that CRM:
1) Is the only method to improve over cross-entropy under average mistake severity (the metric from prior work they argue is flawed)
2) Beats all other metrics under their new, improved metric
3) Improves on the calibration of the predictions

Strengths:
The method is very simple and easy to understand, builds upon existing work, and I could probably implement it from scratch in 30 minutes on top of my existing models. While ML reviewers very frequently about a lack of complexity, I think this is a great strength.

The methods section is very clearly written, and it was quite easy to quickly understand what the method is doing.

Weaknesses:
I am very concerned about the experiment sections.

1)
To my understanding, Figure 2/Section 4.1 are factually incorrect. In particular, it appears that the soft-labels technique does essentially the same, or better than, CRM, across all fronts. In detail,
a)
In Figure 2(a), the leftmost softlabel point is equal to or better than CRM (and cross-entropy)

b)
Figure 2(b) really concerns me, as it appears that the hyperparameters have been chosen to make a fairly narrow point - that it is possible to have low average hierarchical distance, and high top-1 error. I agree that that indicates a problem with the metric.

However, the authors make the fair broader claim that CRM is the only method which beats cross-entropy, which I do not think is justified. Looking at Figure 4 in A.1, it is readily apparent that choosing different hyperparameters for existing methods would yield similar error distributions to CRM. Having these plots in the appendix, combined with claims that the authors chose the best hyperparameters, feels a bit misleading.

c)
As in 1), soft labels is essentially on top of CRM and Cross entropy (for iNaturalist19, it looks like a higher beta value would be directly on top, it's unclear why the authors did not extend the curve further)

2)
These results, at first blush, seem fairly impressive. For the leftmost plots, I am concerned that the authors are using subpar hyperparameters, similarly to 1)(b) above. Strangely, in this instance the results for other hyperparameters are not included in the appendix. The remaining experiments are fairly convincing, though.

Reccomendation
I do not think this paper can be accepted in its current form. While I suspect that CRM is a good method that I would like to use, some of the core arguments (Figure 2/Section 4.1) in the paper appear to be fatally flawed.

Smaller notes:
- The paper could use an additional proofread, as there are often odd phrasings. I found the experiment section particularly hard to follow
- The acronym HXE is never defined, or linked to a citation

---

> ### Author Response · Authors · 2020-11-19
> **Resolving confusion about Figure 2 and hyperparameters. Glad they found simplicity as one of the major strengths of our approach.**
>
> We thank the reviewer for their feedback. We also thank the reviewer for finding simplicity as one of the major strengths of our approach (CRM) as it does not require retraining, hyperparameter cross-validation, and can be used with almost no computational overhead.
>
> The major questions/concerns raised by the reviewer primarily are related to Figure 2, evaluation metric, and hyperparameters. Below we discuss them all in detail and hope to resolve all the concerns. We hope the reviewer makes an additional pass over the experiment section after resolving the confusion for better clarity.
>
> Figure 2
>
> Performance: Note that the purpose of Figure 2 is to show how current metrics to evaluate mistake severity do not reflect the true nature of the models. We do this by highlighting that the current evaluation of mistake severity, which is evaluated _only_ over the incorrectly classified samples (hence _different_ test sets for different models as the mistakes will be different), will favour models making additional _easy mistakes_ (as it involves division by the number of mistakes) which is undesirable. We propose an easy fix to correct this bias which involves evaluation of mistake severity on a _fixed_ test set (not only on the misclassified ones), and, as expected, this simple fix changes the evaluation significantly and provides a reliable evaluation.
>
> Now, looking at Fig 2(a) (the old way of evaluating only on mistakes), it seems like softlabels with $\beta = 4$ and HXE with $\alpha=0.6$ are the best models with the lowest hierarchical distance@1 over mistakes (average mistake severity) as seen on the bottom right. However, as we correct the above said bias, those very models show the highest hierarchical distance@1 (Fig 2(c) top right). And to answer why this is the case, the mistake distribution in Fig 2(b) clearly shows that this model is just making more easy mistakes while *not* improving at all the hierarchical mistakes over the entire test set.
>
> Hyperparameters: We *did not* cherry-pick hyperparameters $\alpha$ and $\beta$ in Fig 2(b). They are the best-performing ones (refer [2]). In fact, we provided a proper analysis to show the impact of hyperparameter on the mistake distribution (Figure 4 in A.1). In the case of softlabels, as $\beta$ decreases, the model is supposed to become better aligned towards the hierarchy with a tradeoff with top1 accuracy, but in reality, performance simply gets worse. Note, with increasing values of $\beta$, the softlabel method becomes similar to cross-entropy. Practically, beyond $\beta=50$ its performance converges, remaining almost the same. To make it more evident, we added four additional points corresponding to $\beta$ = [50,75,100,200] in Figure 2 for iNaturalist19. As expected, they all converge to a point (refer Table2 in A.4 for exact values).
>
> Therefore, at high values of $\beta$, softlabel is very similar to cross-entropy; otherwise, it merely adds easy mistakes. Please note, this observation is for hierarchical distance@1. As we have shown in Figure 3, softlabel performs better than cross-entropy for ranking classes measured by distance@k, but still worse than CRM.
>
> > For the leftmost plots (in Fig 3), I am concerned that the authors are using subpar hyperparameters.
>
> Regarding Figure 3, for a fair comparison, we picked the hyperparameters for other models where their best results occurred. We have added results with all other hyperparameters in the appendix (Fig 5&6) to justify this claim: We can observe that the best results mostly occur at $\beta=4$ and $\alpha=0.6$. We hope this addresses the reviewer’s concern and shows that our method outperforms existing methods at their optimal parameters (y-axis is in log-scale w.r.t classes) both in terms of ranking classes in Fig 3 (left) and tradeoffs between hierarchical distance@k and accuracy in Fig 3 (centre and right).
>
> > The authors make the fair broader claim that CRM is the only method which beats cross-entropy, which I do not think is justified:
>
> We have rephrased the suggested line in Section 4.1 to better align with our intended claim (as stated in abstract & introduction)-- the existing methods do not practically improve over the cross-entropy baseline when considering the top-1 hierarchical distance.
>
> > Odd phrasings, missing citations. HXE acronym
>
> Thanks for pointing this out! We have fixed these in the latest draft for ensuring better clarity for readers.
>
> [2] Bertinetto et al., Making Better Mistakes: Leveraging Class Hierarchies with Deep Networks, CVPR20

---

> > ### Comment · AnonReviewer3 · 2020-11-21
> > **My bad**
> >
> > I'm increasing my score from 4 -> 6
> >
> > I thought the empirical arguments in the paper were broken, but in reality I misunderstood, and the arguments were not communicated in a clear way.
> >
> > The discussion around Figure 2 originally contained statements arguing that Fig 2 showed the superiority of CRM. It did not, and does not. The author's response (and returning to the paper a second time) made me realize that.
> >
> > I'm now reasonably convinced the introduced method is a useful method that I'd want to use, if faced with this problem. Based on methods alone, I'd score a 7 or 8.
> >
> > However, the communication in the paper is still quite bad, and contains some dodgy statements. For instance, the phrase [1] is just wrong, as it is evaluating other methods under the new metric based on hyperparameters chosen to optimize the old metric. The paper/results section should also be very clearly structured to indicate that there are 2 different but related points being made (that is somewhat unique amongst ICLR papers), e.g. by putting the two points in different sections.
> >
> > [1]" As illustrated in Figure 2(c), existing state-of-the-art hierarchy-aware classification models are worse than the cross-entropy baseline, with a sharp drop in classification accuracy without improving the severity of top-1 mistakes."

---

> > > ### Author Response · Authors · 2020-11-23
> > > **Thanks for active feedback! We made changes according to suggestions, and hope to resolve the clarity aspect**
> > >
> > > We thank AR3 for their active engagement and constructive feedback. We are very happy the confusion was resolved, and they found our method useful for tackling this problem. We agree with the concerns AR3 raises, and hope to improve the writing -- especially in Section 4.1, which could be misunderstood due to confusion about Fig 2.
> > >
> > > We have updated the draft with the following changes:
> > >
> > > > (4.1) badly communicated, dodgy statements
> > >
> > > - We rewrote Section 4.1 entirely to eliminate any such statements, reframing it similar to our rebuttal comment about Fig 2 but with more clear, concise statements which we hope entirely resolves this issue.
> > >
> > > > The paper/results section should also be very clearly structured to indicate that there are 2 different but related points being made
> > >
> > > - We restructured the results subsection by replacing it with three subsections, each  corresponding to a different aspect in hierarchy-aware classification
> > >   - 4.1 being making better top-1 mistakes
> > >   - 4.2 being about ranking classes aka top-k mistakes across k
> > >   - 4.3 talking about calibration aspects of the problem
> > > This structure aligns better with the three points stated in the contributions (introduction).
> > >
> > > We hope this fully resolves their concerns about structure and clarity.

---

### Official Review · AnonReviewer4 · 2020-10-26
**The authors show that previous related studies have used an incomplete metric for evaluation, which is an important finding. But their solution lacks enough novelty.**

**Rating:** 6
**Confidence:** 4

**Review:**

Summary:
The authors propose a model to improve the output distribution of neural nets in image classification problems. Their model is a post hoc procedure and is based on the tree structure of WordNet. The model revises the classifier output based on the distance of the labels in the tree. Intuitively, their solution is to pick the candidate label that is located in the region of the tree with a higher accumulated probability mass value. They also experimentally show that the previous evaluation metrics are inconclusive.

Pros:
- The authors provide a different perspective on the evaluation procedure of the previous studies and experimentally show that it was incomplete. This is an important finding.
- Their experiments are thorough.

Cons:
- The article lacks enough novelty: The problem has been investigated before. The solution is not novel. The WordNet tree structure has been extensively used in the information retrieval community before.
- The article is not written well: There are informal vocabulary in the paper (e.g., “something similar” or “grossly miscalibrated”). There are also typos (e.g., see the paragraph before Theorem 1). In Section 3 it is not formally stated that the tree structure is derived from WordNet (the authors mention this in Abstract section). In Section 4 the baselines are not cited!

---

> ### Author Response · Authors · 2020-11-19
> **Request to not dismiss the paper solely because of lack of novelty**
>
> We thank the reviewer for their thoughtful feedback.
>
> > The article lacks enough novelty: The problem has been investigated before. The solution is not novel.
>
> We would like to highlight the following here:
>  - As opposed to all recent literature in hierarchy-aware classification, our approach *does not require retraining*
>  - Previous works often introduce algorithmic novelty by designing ad-hoc loss functions and retraining models on ImageNet with sensitive additional hyperparameters, which we demonstrate do not improve over the cross-entropy baseline in making better mistakes.
> - Our simple method beats all existing sophisticated methods with no retraining in ranking classes on hierarchical distance@k by large margins while preserving far better calibration necessary for real-world use.
> - The novelty we introduce is not in the algorithm itself, but primarily in uncovering important shortcomings in literature, demonstrating a simple, classical approach which works effectively when combined with deep models and use its simplicity to provide insights into the problem (Theorem 1).
>  - We believe our method can have a major real-world impact due to its simplicity and deployability with no additional training.
>
> We request the reviewer to reconsider and not dismiss the paper solely on this ground.
>
> > In Section 3 it is not formally stated that the tree structure is derived from WordNet
>
> The taxonomy need not be from WordNet-- approaches should work with any given hierarchy. We also demonstrate this in experiments with the large-scale dataset iNaturalist-19 which uses the biological taxonomy. We have added this in Section 3 for better clarity.
>
> > informal vocabulary, typos and missing citations.
>
> Thanks for pointing this out! We have corrected informal vocabulary, added detail to make crisper statements, fixed all typos and missing citations issues in the updated draft. Hope it resolves this issue.

---

> > ### Comment · AnonReviewer4 · 2020-11-21
> > **Score update for improving the presentation**
> >
> > Thanks for the clarifications and also for editing your paper.
> >
> > I am updating my score from 4 to 6 (a border line paper), because:
> > - The presentation has improved.
> > - I personally would like to see a clear technical novelty in the top venue papers. While I agree that the proposed method is simple, post-hoc, and methodical, I also keep reminding myself that it is not the authors method! It was published before. In my opinion applying an old technique is acceptable only if either it is introduced from a distinctly different research area or it results in a substantial improvement in the same area (this can happen for various reasons). I believe the paper is a border line paper because it describes a problem in the evaluation process and it helps the future studies to avoid this issue again.

---

### Official Review · AnonReviewer1 · 2020-10-28
**A straightforward technique with satisfying performance**

**Rating:** 7
**Confidence:** 2

**Review:**

This paper proposes to use conditional risk minimization (CRM) for hierarchy aware classification. The proposed method simply amends mistakes using a cost matrix with the lowest common ancestor information. The method outperforms SOTA deep hierarchy-aware classifiers by large margins at ranking classes with little loss in classification accuracy.

As the authors mentioned, CRM was already proposed several decades ago, so the novelty of this paper is limited. However, the paper does demonstrate the power of this old technique when equipped with modern deep learning tools. Also, the simplicity and intuition are much appreciated.

I myself am not an expert in image classification. However, for text classification, recent studies (e.g., [1]) shows that using binary cross-entropy (i.e., viewing a multi-label classification problem as L binary classification task, where L is the number of classes) can achieve higher performance than multi-label cross-entropy. In this case, it is possible that more than one label can have p(y|x) > 0.5. I wonder whether your approach is still effective for models using binary cross-entropy. It would also be better if the authors can show some experimental results on hierarchical text classification.

[1] Liu et al. Deep Learning for Extreme Multi-label Text Classification. SIGIR'17.

---

> ### Author Response · Authors · 2020-11-19
> **Thanks for the positive feedback. Glad they found CRM combined with modern deep learning tools impactful in its simplicity and intuition**
>
> We thank the reviewer for their positive feedback and are glad that they found our demonstration of CRM combined with modern deep learning tools impactful in its simplicity and intuition.
>
> > I wonder whether your approach is still effective for (multi-label) models using binary cross-entropy
>
> We believe it would not be trivial to directly apply CRM, or any recent label-hierarchy approach directly with binary cross-entropy since there is no straightforward way to combine L sigmoids into one so that different probabilities can be compared. We did not think about this particular problem as the community at large is mostly focused on single-label classification; however, it definitely is an interesting question, and we would like to explore it properly in the future. Thank you for this proposal.
>
> > It would also be better if the authors can show some experimental results on hierarchical text classification.
>
> Could the reviewer point to a suitable hierarchical multi-class text classification datasets with an elaborate hierarchy (>3 depth)? We will experiment using cross-entropy and CRM corrections on them and do our best to post our findings if available before the closure of the discussion phase otherwise will add them in the draft.

---

> > ### Comment · AnonReviewer1 · 2020-11-24
> > **Thanks for the clarifications**
> >
> > Thanks for the clarifications. You might try the RCV1 dataset (103 classes, depth 6) if you would like to add some results on hierarchical text classification. I will keep my score.

---

### Official Review · AnonReviewer2 · 2020-10-30
**A nice paper that thoroughly tests CRM for hierarchical classification using latest NNs**

**Rating:** 8
**Confidence:** 4

**Review:**

The paper addresses hierarchical classification, where the classes live in a hierarchy, and the cost of a mistake is the tree distance between the nodes.

The paper tests the latest cool algorithms for hierarchical-aware loss functions, versus a very old idea: CRM. In CRM, you make your best estimate of the posterior probability of a class y given input x P(y|x), and then you make a final decision based on minimizing the expected loss.

There seems to be a belief that modifying the loss function to be hierarchy-aware is clearly better than doing boring old CRM. But there is not much evidence in favor of that hypothesis. This paper offers negative evidence for that hypothesis, with two experiments:

1.  By comparing hierarchical loss to top-1 loss with modified loss functions, there is a tradeoff, and there does not seem to be an advantage in using the modified loss function.
2.  For the top-k case, using CRM clearly dominates the proposals for modifying the loss function.

These support the use of CRM.

I find this paper to be really nice -- I'd far rather have a paper with good experiments with known algorithms, where I can learn something useful; than a paper with a new algorithm with somewhat useless experiments. So I would argue for acceptance.

One thing for the authors to think about:

When they test the calibration of the modified loss functions, they find them to be poorly calibrated. This is not surprising, since the modified loss functions are not proper scoring rules. They attempt to calibrate by using a softmax  with variable T. Wouldn't it make more sense to train exp(alpha_i x + beta_i) / \sum_{i=1}^N exp(alpha_i x + beta_i) ? that is, a gain and offset for all classes after the first one?

---

> ### Author Response · Authors · 2020-11-19
> **Thanks for the encouraging feedback. Glad they found our paper to be insightful and our experiments thorough**
>
> We thank the reviewer for their positive feedback.
>
> > Wouldn't it make more sense to train exp(alpha_i x + beta_i) / \sum_{i=1}^N exp(alpha_i x + beta_i) instead of T?
>
> We agree that there are many more sophisticated approaches to improve calibration; however since our focus was to show the effectiveness of CRM. We used temperature scaling as it has only 1 degree of freedom and does not affect the ranking of the classes or accuracy. The suggested approach, which is similar to vector scaling [1], and similarly more sophisticated ways to enforce calibration have an important issue: they are expected to significantly alter the weighting between different classes which might have an unintended effect on the hierarchical corrections made and impact their decisions-- hence we avoid them.
>
> We are nevertheless performing experiments as requested based on this suggestion and most likely post our findings within the next two days.
>
> [1] Guo et al. On Calibration of Modern Neural Networks, ICML17

---

> > ### Author Response · Authors · 2020-11-23
> > **Vector scaling results**
> >
> > We trained 2 vectors  using this equation exp(alpha_i x + beta_i) / \sum_{i=1}^N exp(alpha_i x + beta_i) which modify the logits in a post-hoc manner. This is done by minimizing the Negative Log Likelihood (NLL) loss on the validation set. We used the Adam optimizer with a learning rate of 0.1 and trained it for 10 epochs. The model is then evaluated on the test set.
> >
> > **Tiered-ImageNet**
> >
> > | Model |  ECE(Temp.)  | ECE(Vector) | Accuracy(Temp.) | Accuracy(Vector) |  Hier Dist@5(Temp.) | Hier Dist@5(Vector) | Hier Dist@20(Temp.) | Hier Dist@20(Vector) |
> > | :------------------- |:----------|:------------|:------------|:-------------|:------|:------------|:------------|:-------------|
> > | Cross-Entropy        | 1.61      | 1.35        | 68.76        | 69.03         | 5.64  | 5.64        | 7.08        | 7.08         |
> > | Soft-Labels(Beta=4)  | 11.12     | 6.74        | 58.78        | 48.21         | 5.19  | 5.36        | 6.26        | 6.40         |
> > | Soft-Labels(Beta=5)  | 10.92     | 7.83        | 62.78        | 56.26         | 5.20  | 5.28        | 6.28        | 6.36         |
> > | Soft-Labels(Beta=10) | 6.36      | 7.24        | 67.63        | 67.28         | 5.42  | 5.44        | 6.60        | 6.65         |
> > | Soft-Labels(Beta=15) | 4.79      | 2.74        | 68.88        | 69.13         | 5.55  | 5.55        | 6.84        | 6.87         |
> > | HXE(Alpha=0.2)       | 1.53      | 1.56        | 68.23        | 68.51         | 5.54  | 5.54        | 6.91        | 6.92         |
> > | HXE(Alpha=0.4)       | 2.44      | 2.37        | 65.88        | 65.55         | 5.48  | 5.47        | 6.79        | 6.80         |
> > | HXE(Alpha=0.5)       | 2.61      | 4.01        | 64.22        | 64.00         | 5.46  | 5.46        | 6.75        | 6.76         |
> > | HXE(Alpha=0.6)       | 3.28      | 6.19        | 57.70        | 57.63         | 5.47  | 5.47        | 6.67        | 6.68         |
> >
> > **iNaturalist19**
> >
> > | Model |  ECE(Temp.)  | ECE(Vector) | Accuracy(Temp.) | Accuracy(Vector) |  Hier Dist@5(Temp.) | Hier Dist@5(Vector) | Hier Dist@20(Temp.) | Hier Dist@20(Vector) |
> > | :------------------- |:----------|:------------|:------------|:-------------|:------|:------------|:------------|:-------------|
> > | Cross-Entropy        | 1.42      | 2.01        | 59.74        | 59.22         | 1.83  | 1.84        | 2.85        | 2.86         |
> > | Soft-Labels(Beta=4)  | 11.46     | 5.73        | 20.18        | 9.18          | 1.55  | 1.63        | 2.00        | 2.11         |
> > | Soft-Labels(Beta=5)  | 17.20     | 5.75        | 26.10        | 13.92         | 1.50  | 1.54        | 1.97        | 2.04         |
> > | Soft-Labels(Beta=10) | 19.87     | 26.55       | 46.61        | 42.24         | 1.41  | 1.43        | 1.94        | 1.96         |
> > | Soft-Labels(Beta=15) | 16.63     | 24.31       | 53.28        | 50.53         | 1.42  | 1.43        | 1.98        | 1.99         |
> > | HXE(Alpha=0.2)       | 1.50      | 0.58        | 58.50        | 58.25         | 1.66  | 1.65        | 2.53        | 2.54         |
> > | HXE(Alpha=0.4)       | 1.13      | 2.89        | 54.29        | 53.60         | 1.57  | 1.57        | 2.33        | 2.33         |
> > | HXE(Alpha=0.5)       | 2.46      | 5.21        | 50.40        | 49.50         | 1.56  | 1.56        | 2.27        | 2.26         |
> > | HXE(Alpha=0.6)       | 5.24      | 5.99        | 44.34        | 42.97         | 1.58  | 1.58        | 2.24        | 2.24         |
> >
> >
> > We observe from the table that:
> >  - As predicted, Softlabels beta=4/5 get better calibrated (far lower ECE) but the hierarchical corrections get undone -- regressing significantly on all performance metrics-- accuracy and average hierarchical dist@5/20 significantly.
> >  - In other cases there isn't a significant impact on the hierarchical corrections since the change in calibration is slight. However, similar to observations in Guo etal [1] -- vector scaling does not necessarily achieve better calibration compared to temperature scaling in all cases, it often overfits to the validation set.

---

### Decision · Program_Chairs · 2021-01-07
**Final Decision**

**Decision:**

Accept (Poster)

**Comment:**

The approach explore the use of Conditional Risk Minimization (CRM) as a post-hoc operation to amend a classifier decision by averaging a prior class hierarchy. The authors show that it is beneficial for ranking predictions without sacrifying top-1 accuracy.

The rebuttal period clarified some reviewers' concern on paper presentation and experiments, and all reviewers recommend acceptance after the discussion period.

Although the approach is simple and directly revisits the use of CRM for deep models, the AC considers that the contribution is meaningful, and that the proposed method provides predictions with good ranking and calibration properties. The paper also sheds light into interesting issues in state-of-the-art methods integrating class hierarchies during training.
The AC therefore recommends paper acceptance.